# Deterministic Approximation for Submodular Maximization over a Matroid in Nearly Linear Time

**Kai Han    Zongmai Cao    Shuang Cui    Benwei Wu**
School of Computer Science and Technology / Suzhou Research Institute
University of Science and Technology of China
hankai@ustc.edu.cn, {czm18,lakers,wubenwei}@mail.ustc.edu.cn

## Abstract

We study the problem of maximizing a non-monotone, non-negative submodular function subject to a matroid constraint. The prior best-known deterministic approximation ratio for this problem is $\frac{1}{4} - \epsilon$ under $\mathcal{O}((n^4/\epsilon)\log n)$ time complexity. We show that this deterministic ratio can be improved to $\frac{1}{4}$ under $\mathcal{O}(nr)$ time complexity, and then present a more practical algorithm dubbed TwinGreedyFast which achieves $\frac{1}{4} - \epsilon$ deterministic ratio in nearly-linear running time of $\mathcal{O}(\frac{n}{\epsilon}\log\frac{r}{\epsilon})$. Our approach is based on a novel algorithmic framework of simultaneously constructing two candidate solution sets through greedy search, which enables us to get improved performance bounds by fully exploiting the properties of independence systems. As a byproduct of this framework, we also show that TwinGreedyFast achieves $\frac{1}{2p+2} - \epsilon$ deterministic ratio under a $p$-set system constraint with the same time complexity. To showcase the practicality of our approach, we empirically evaluated the performance of TwinGreedyFast on two network applications, and observed that it outperforms the state-of-the-art deterministic and randomized algorithms with efficient implementations for our problem.

## 1  Introduction

Submodular function maximization has aroused great interests from both academic and industrial societies due to its wide applications such as crowdsourcing [47], information gathering [35], sensor placement [33], influence maximization [37, 48] and exemplar-based clustering [30]. Due to the large volume of data and heterogeneous application scenarios in practice, there is a growing demand for designing accurate and efficient submodular maximization algorithms subject to various constraints.

Matroid is an important structure in combinatorial optimization that abstracts and generalizes the notion of linear independence in vector spaces [8]. The problem of submodular maximization subject to a matroid constraint (SMM) has attracted considerable attention since the 1970s. When the considered submodular function $f(\cdot)$ is monotone, the classical work of Fisher et al. [27] presents a deterministic approximation ratio of $1/2$, which keeps as the best deterministic ratio during decades until Buchbinder et al. [14] improve this ratio to $1/2 + \epsilon$ very recently.

When the submodular function $f(\cdot)$ is non-monotone, the best-known deterministic ratio for the SMM problem is $1/4 - \epsilon$, proposed by Lee et al. [38], but with a high time complexity of $\mathcal{O}((n^4\log n)/\epsilon)$. Recently, there appear studies aiming at designing more efficient and practical algorithms for this problem. In this line of studies, the elegant work of Mirzasoleiman et al. [44] and Feldman et al. [25] proposes the best deterministic ratio of $1/6 - \epsilon$ and the fastest implementation with an expected ratio of $1/4$, and their algorithms can also handle more general constraints such as a $p$-set system constraint. For clarity, we list the performance bounds of these work in Table 1. However, it is still unclear whether the $1/4 - \epsilon$ deterministic ratio for a single matroid constraint can be further improved, or whether there exist faster algorithms achieving the same $1/4 - \epsilon$ deterministic ratio.

Table 1: Approximation for Non-monotone Submodular Maximization over a Matroid

| Algorithms | Ratio | Time Complexity | Type |
|---|---|---|---|
| Lee et al. [38] | $1/4 - \epsilon$ | $\mathcal{O}((n^4 \log n)/\epsilon)$ | Deterministic |
| Mirzasoleiman et al. [44] | $1/6 - \epsilon$ | $\mathcal{O}(nr + r/\epsilon)$ | Deterministic |
| Feldman et al. [25] | $1/4$ | $\mathcal{O}(nr)$ | Randomized |
| Buchbinder and Feldman [10] | $0.385$ | $\text{poly}(n)$ | Randomized |
| TwinGreedy (Alg. 1) | $1/4$ | $\mathcal{O}(nr)$ | Deterministic |
| TwinGreedyFast (Alg. 2) | $1/4 - \epsilon$ | $\mathcal{O}((n/\epsilon) \log(r/\epsilon))$ | Deterministic |

In this paper, we propose an approximation algorithm TwinGreedy (Alg. 1) with a deterministic $1/4$ ratio and $\mathcal{O}(nr)$ running time for maximizing a non-monotone, non-negative submodular function subject to a matroid constraint, thus improving the best-known $1/4 - \epsilon$ deterministic ratio of Lee et al. [38]. Furthermore, we show that the solution framework of TwinGreedy can be implemented in a more efficient way, and present a new algorithm dubbed TwinGreedyFast with $1/4 - \epsilon$ deterministic ratio and nearly-linear $\mathcal{O}(\frac{n}{\epsilon} \log \frac{r}{\epsilon})$ running time. To the best of our knowledge, TwinGreedyFast is the fastest algorithm achieving the $\frac{1}{4} - \epsilon$ deterministic ratio for our problem in the literature. As a byproduct, we also show that TwinGreedyFast can be used to address a more general $p$-set system constraint and achieves a $\frac{1}{2p+2} - \epsilon$ approximation ratio with the same time complexity.

It is noted that most of the current deterministic algorithms for non-monontone submodular maximization (e.g., [25, 44, 45]) leverage the "repeated greedy-search" framework proposed by Gupta et al. [31], where two or more candidate solution sets are constructed successively and then an unconstrained submodular maximization (USM) algorithm (e.g., [12]) is called to find a good solution among the candidate sets and their subsets. Our approach is based on a novel "simultaneous greedy-search" framework different from theirs, where two disjoint candidate solution sets $S_1$ and $S_2$ are built simultaneously with only single-pass greedy searching, without calling a USM algorithm. We call these two solution sets $S_1$ and $S_2$ as "twin sets" because they "grow up" simultaneously. Thanks to this framework, we are able to bound the "utility loss" caused by greedy searching using $S_1$ and $S_2$ themselves, through a careful classification of the elements in an optimal solution $O$ and mapping them to the elements in $S_1 \cup S_2$. Furthermore, by incorporating a thresholding method inspired by Badanidiyuru and Vondrák [2] into our framework, the TwinGreedyFast algorithm achieves nearly-linear time complexity by only accepting elements whose marginal gains are no smaller than given thresholds.

We evaluate the performance of TwinGreedyFast on two applications: social network monitoring and multi-product viral marketing. The experimental results show that TwinGreedyFast runs more than an order of magnitude faster than the state-of-the-art efficient algorithms for our problem, and also achieves better solution quality than the currently fastest randomized algorithms in the literature.

## 1.1 Related Work

When the considered submodular function $f(\cdot)$ is monotone, Calinescu et al. [15] propose an optimal $1 - 1/e$ expected ratio for the problem of submodular maximization subject to a matroid constraint (SMM). The SMM problem seems to be harder when $f(\cdot)$ is non-monotone, and the current best-known expected ratio is $0.385$ [10], got after a series of studies [24, 29, 38, 50]. However, all these approaches are based on tools with high time complexity such as multilinear extension.

There also exist efficient deterministic algorithms for the SMM problem: Gupta et al. [31] are the first to apply the "repeated greedy search" framework described in last section and achieve $1/12 - \epsilon$ ratio, which is improved to $1/6 - \epsilon$ by Mirzasoleiman et al. [44] and Feldman et al. [25]; under a more general $p$-set system constraint, Mirzasoleiman et al. [44] achieve $\frac{p}{(p+1)(2p+1)} - \epsilon$ deterministic ratio and Feldman et al. [25] achieve $\frac{1}{p+2\sqrt{p+3}} - \epsilon$ deterministic ratio (assuming that they use the USM algorithm with $1/2 - \epsilon$ deterministic ratio in [9]); some studies also propose streaming algorithms under various constraints [32, 45].

As regards the efficient randomized algorithms for the SMM problem, the SampleGreedy algorithm in [25] achieves $1/4$ expected ratio with $\mathcal{O}(nr)$ running time; the algorithms in [11] also achieve

a $1/4$ expected ratio with slightly worse time complexity of $\mathcal{O}(nr \log n)$ and a $0.283$ expected ratio under cubic time complexity of $\mathcal{O}(nr \log n + r^{3+\epsilon})$[1]; Chekuri and Quanrud [16] provide a $0.172 - \epsilon$ expected ratio under $\mathcal{O}(\log n \log r/\epsilon^2)$ adaptive rounds; and Feldman et al. [26] propose a $1/(3 + 2\sqrt{2})$ expected ratio under the steaming setting. It is also noted that Buchbinder et al. [14] provide a de-randomized version of the algorithm in [11] for monotone submodular maximization, which has time complexity of $\mathcal{O}(nr^2)$. However, it remains an open problem to find the approximation ratio of this de-randomized algorithm for the SMM problem with a non-monotone objective function.

A lot of elegant studies provide efficient submodular optimization algorithms for monotone submodular functions or for a cardinality constraint [2, 4, 5, 13, 22, 23, 36, 43]. However, these studies have not addressed the problem of non-monotone submodular maximization subject to a general matroid (or $p$-set system) constraint, and our main techniques are essentially different from theirs.

## 2   Preliminaries

Given a ground set $\mathcal{N}$ with $|\mathcal{N}| = n$, a function $f : 2^{\mathcal{N}} \mapsto \mathbb{R}$ is submodular if for all $X, Y \subseteq \mathcal{N}$, $f(X) + f(Y) \geq f(X \cup Y) + f(X \cap Y)$. The function $f(\cdot)$ is called non-negative if $f(X) \geq 0$ for all $X \subseteq \mathcal{N}$, and $f(\cdot)$ is called non-monotone if $\exists X \subset Y \subseteq \mathcal{N} : f(X) > f(Y)$. For brevity, we use $f(X \mid Y)$ to denote $f(X \cup Y) - f(Y)$ for any $X, Y \subseteq \mathcal{N}$, and write $f(\{x\} \mid Y)$ as $f(x \mid Y)$ for any $x \in \mathcal{N}$. We call $f(X \mid Y)$ as the "marginal gain" of $X$ with respect to $Y$.

An independence system $(\mathcal{N}, \mathcal{I})$ consists of a finite ground set $\mathcal{N}$ and a family of independent sets $\mathcal{I} \subseteq 2^{\mathcal{N}}$ satisfying: (1) $\emptyset \in \mathcal{I}$; (2) If $A \subseteq B \in \mathcal{I}$, then $A \in \mathcal{I}$ (called hereditary property). An independence system $(\mathcal{N}, \mathcal{I})$ is called a matroid if it satisfies: for any $A \in \mathcal{I}, B \in \mathcal{I}$ and $|A| < |B|$, there exists $x \in B \backslash A$ such that $A \cup \{x\} \in \mathcal{I}$ (called exchange property).

Given an independence system $(\mathcal{N}, \mathcal{I})$ and any $X \subseteq Y \subseteq \mathcal{N}$, $X$ is called a base of $Y$ if: (1) $X \in \mathcal{I}$; (2) $\forall x \in Y \backslash X : X \cup \{x\} \notin \mathcal{I}$. The independence system $(\mathcal{N}, \mathcal{I})$ is called a $p$-set system if for every $Y \subseteq \mathcal{N}$ and any two basis $X_1, X_2$ of $Y$, we have $|X_1| \leq p|X_2|$ ($p \geq 1$). It is known that $p$-set system is a generalization of several structures on independence systems including matroid, $p$-matchoid and $p$-extendible system, and an inclusion hierarchy of these structures can be found in [32].

In this paper, we consider a non-monotone, non-negative submodular function $f(\cdot)$. Given $f(\cdot)$ and a matroid $(\mathcal{N}, \mathcal{I})$, our optimization problem is $\max\{f(S) : S \in \mathcal{I}\}$. In the end of Section 4, we will also consider a more general case where $(\mathcal{N}, \mathcal{I})$ is a $p$-set system.

We introduce some frequently used properties of submodular functions. For any $X \subseteq Y \subseteq \mathcal{N}$ and any $Z \subseteq \mathcal{N} \backslash Y$, we have $f(Z \mid Y) \leq f(Z \mid X)$; this property can be derived by the definition of submodular functions. For any $X, Y \subseteq \mathcal{N}$ and a partition $Z_1, Z_2, \cdots, Z_t$ of $Y \backslash X$, we have

$$f(Y \mid X) = \sum_{j=1}^{t} f(Z_j \mid Z_1 \cup \cdots \cup Z_{j-1} \cup X) \leq \sum_{j=1}^{t} f(Z_j \mid X) \tag{1}$$

For convenience, we use $[h]$ to denote $\{1, \cdots, h\}$ for any positive integer $h$, and use $r$ to denote the *rank* of $(\mathcal{N}, \mathcal{I})$, i.e., $r = \max\{|S| : S \in \mathcal{I}\}$, and denote an optimal solution to our problem by $O$.

## 3   The TwinGreedy Algorithm

In this section, we consider a matroid constraint and introduce the TwinGreedy algorithm (Alg. 1) that achieves $1/4$ approximation ratio. The TwinGreedy algorithm maintains two solution sets $S_1$ and $S_2$ that are initialized to empty sets. At each iteration, it considers all the candidate elements in $\mathcal{N} \backslash (S_1 \cup S_2)$ that can be added into $S_1$ or $S_2$ without violating the feasibility of $\mathcal{I}$. If there exists such an element, then it greedily selects $(e, S_i)$ where $i \in \{1, 2\}$ such that adding $e$ into $S_i$ can bring the maximal marginal gain $f(e \mid S_i)$ without violating the feasibility of $\mathcal{I}$. TwinGreedy terminates when no more elements can be added into $S_1$ or $S_2$ with a positive marginal gain while still keeping the feasibility of $\mathcal{I}$. Then it returns the one between $S_1$ and $S_2$ with the larger objective function value.

**Algorithm 1:** TwinGreedy($\mathcal{N}, \mathcal{I}, f(\cdot)$)

1   $S_1 \leftarrow \emptyset; S_2 \leftarrow \emptyset;$
2   **repeat**
3      $\mathcal{M}_1 \leftarrow \{e \in \mathcal{N} \backslash (S_1 \cup S_2) : S_1 \cup \{e\} \in \mathcal{I}\}$
4      $\mathcal{M}_2 \leftarrow \{e \in \mathcal{N} \backslash (S_1 \cup S_2) : S_2 \cup \{e\} \in \mathcal{I}\}$
5      $C \leftarrow \{j \mid j \in \{1, 2\} \wedge \mathcal{M}_j \neq \emptyset\}$
6      **if** $C \neq \emptyset$ **then**
7         $(i, e) \leftarrow \arg\max_{j \in C, u \in \mathcal{M}_j} f(u \mid S_j);$ (ties broken arbitrarily)
8         **if** $f(e \mid S_i) \leq 0$ **then Break**;
9         $S_i \leftarrow S_i \cup \{e\};$
10   **until** $\mathcal{M}_1 \cup \mathcal{M}_2 = \emptyset;$
11   $S^* \leftarrow \arg\max_{X \in \{S_1, S_2\}} f(X)$
12   **return** $S^*$

Although the TwinGreedy algorithm is simple, its performance analysis is highly non-trivial. Roughly speaking, as TwinGreedy adopts a greedy strategy to select elements, we try to find some "competitive relationships" between the elements in $O$ and those in $S_1 \cup S_2$, such that the total marginal gains of the elements in $O$ with respect to $S_1$ and $S_2$ can be upper-bounded by $S_1$ and $S_2$'s own objective function values. However, this is non-trivial due to the correlation between the elements in $S_1$ and $S_2$. To overcome this hurdle, we first classify the elements in $O$, as shown in Definition 1:

**Definition 1** *Consider the two solution sets $S_1$ and $S_2$ when TwinGreedy returns. We can write $S_1 \cup S_2$ as $\{v_1, v_2, \cdots, v_k\}$ where $k = |S_1 \cup S_2|$, such that $v_t$ is added into $S_1 \cup S_2$ by the algorithm before $v_s$ for any $1 \leq t < s \leq k$. With this ordered list, given any $e = v_j \in S_1 \cup S_2$, we define*

$$\mathrm{Pre}(e, S_1) = \{v_1, \cdots, v_{j-1}\} \cap S_1; \quad \mathrm{Pre}(e, S_2) = \{v_1, \cdots, v_{j-1}\} \cap S_2; \qquad (2)$$

*That is, $\mathrm{Pre}(e, S_i)$ denotes the set of elements in $S_i$ ($i \in \{1, 2\}$) that are added by the TwinGreedy algorithm before adding $e$. Furthermore, we define*

$$O_1^+ = \{e \in O \cap S_1 : \mathrm{Pre}(e, S_2) \cup \{e\} \in \mathcal{I}\}; \qquad O_1^- = \{e \in O \cap S_1 : \mathrm{Pre}(e, S_2) \cup \{e\} \notin \mathcal{I}\}$$
$$O_2^+ = \{e \in O \cap S_2 : \mathrm{Pre}(e, S_1) \cup \{e\} \in \mathcal{I}\}; \qquad O_2^- = \{e \in O \cap S_2 : \mathrm{Pre}(e, S_1) \cup \{e\} \notin \mathcal{I}\}$$
$$O_3 = \{e \in O \backslash (S_1 \cup S_2) : S_1 \cup \{e\} \notin \mathcal{I}\}; \qquad O_4 = \{e \in O \backslash (S_1 \cup S_2) : S_2 \cup \{e\} \notin \mathcal{I}\}$$

*We also define the marginal gain of any $e \in S_1 \cup S_2$ as $\delta(e) = f(e \mid \mathrm{Pre}(e, S_1)) \cdot \mathbf{1}_{S_1}(e) + f(e \mid \mathrm{Pre}(e, S_2)) \cdot \mathbf{1}_{S_2}(e)$, where $\mathbf{1}_{S_i}(e) = 1$ if $e \in S_i$ and $\mathbf{1}_{S_i}(e) = 0$ otherwise ($\forall i \in \{1, 2\}$).*

Intuitively, each element $e \in O_1^+ \subseteq S_1$ can also be added into $S_2$ without violating the feasibility of $\mathcal{I}$ when $e$ is added into $S_1$, while the elements in $O_1^- \subseteq S_1$ do not have this nice property. The sets $O_2^+$ and $O_2^-$ can be understood similarly. With the above classification of the elements in $O$, we further consider two groups of elements: $O_1^+ \cup O_1^- \cup O_2^- \cup O_3$ and $O_1^- \cup O_2^+ \cup O_2^- \cup O_4$. By leveraging the properties of independence systems, we can map the first group to $S_1$ and the second group to $S_2$, as shown by Lemma 1 (the proof can be found in the supplementary file). Intuitively, Lemma 1 holds due to the exchange property of matroids.

**Lemma 1** *There exists an injective function $\pi_1 : O_1^+ \cup O_1^- \cup O_2^- \cup O_3 \mapsto S_1$ such that:*

     *1. For any $e \in O_1^+ \cup O_1^- \cup O_2^- \cup O_3$, we have $\mathrm{Pre}(\pi_1(e), S_1) \cup \{e\} \in \mathcal{I}$.*

     *2. For each $e \in O_1^+ \cup O_1^-$, we have $\pi_1(e) = e$.*

*Similarly, there exists an injective function $\pi_2 : O_1^- \cup O_2^+ \cup O_2^- \cup O_4 \mapsto S_2$ such that $\mathrm{Pre}(\pi_2(e), S_2) \cup \{e\} \in \mathcal{I}$ for each $e \in O_1^- \cup O_2^+ \cup O_2^- \cup O_4$ and $\pi_2(e) = e$ for each $e \in O_2^+ \cup O_2^-$.*

The first property shown in Lemma 1 implies that, at the moment that $\pi_1(e)$ is added into $S_1$, $e$ can also be added into $S_1$ without violating the feasibility of $\mathcal{I}$ (for any $e \in O_1^+ \cup O_1^- \cup O_2^- \cup O_3$). This makes it possible to compare the marginal gain of $e$ with respect to $S_1$ with that of $\pi_1(e)$. The

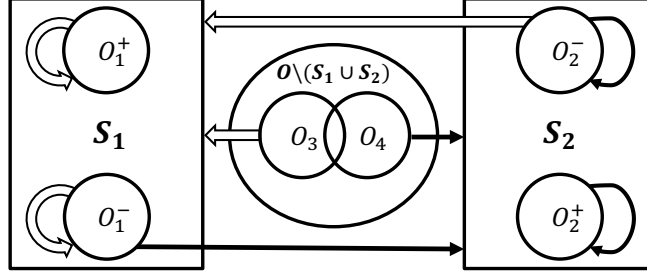

Figure 1: Illustration on the mappings constructed for performance analysis, where the hollow arrows denote $\pi_1(\cdot)$ and the solid arrows denote $\pi_2(\cdot)$.

construction of $\pi_2(\cdot)$ is also based on this intuition. With the two injections $\pi_1(\cdot)$ and $\pi_2(\cdot)$, we can bound the marginal gains of $O_1^+$ to $O_4$ with respect to $S_1$ and $S_2$, as shown by the Lemma 2. Lemma 2 can be proved by using Definition 1, Lemma 1 and the submodularity of $f(\cdot)$.

**Lemma 2** *The TwinGreedy algorithm satisfies:*

$$f(O_1^+ \mid S_2) \le \sum_{e \in O_1^+} \delta(\pi_1(e)); \qquad f(O_2^+ \mid S_1) \le \sum_{e \in O_2^+} \delta(\pi_2(e)) \tag{3}$$

$$f(O_1^- \mid S_2) \le \sum_{e \in O_1^-} \delta(\pi_2(e)); \qquad f(O_2^- \mid S_1) \le \sum_{e \in O_2^-} \delta(\pi_1(e)) \tag{4}$$

$$f(O_4 \mid S_2) \le \sum_{e \in O_4} \delta(\pi_2(e)); \qquad f(O_3 \mid S_1) \le \sum_{e \in O_3} \delta(\pi_1(e)) \tag{5}$$

*where $\pi_1(\cdot)$ and $\pi_2(\cdot)$ are the two functions defined in Lemma 1.*

The proof of Lemma 2 is deferred to the supplementary file. In the next section, we will provide a proof sketch for a similar lemma (i.e., Lemma 3), which can also be used to understand Lemma 2. Now we can prove the performance bounds of TwinGreedy:

**Theorem 1** *When $(\mathcal{N}, \mathcal{I})$ is a matroid, the TwinGreedy algorithm returns a solution $S^*$ with $\frac{1}{4}$ approximation ratio, under time complexity of $\mathcal{O}(nr)$.*

**Proof**: If $S_1 = \emptyset$ or $S_2 = \emptyset$, then we get an optimal solution, which is shown in the supplementary file. So we assume $S_1 \ne \emptyset$ and $S_2 \ne \emptyset$. Let $O_5 = O \backslash (S_1 \cup S_2 \cup O_3)$ and $O_6 = O \backslash (S_1 \cup S_2 \cup O_4)$. By Eqn. (1), we get

$$f(O \cup S_1) - f(S_1) \le f(O_2^+ \mid S_1) + f(O_2^- \mid S_1) + f(O_3 \mid S_1) + f(O_5 \mid S_1) \tag{6}$$

$$f(O \cup S_2) - f(S_2) \le f(O_1^+ \mid S_2) + f(O_1^- \mid S_2) + f(O_4 \mid S_2) + f(O_6 \mid S_2) \tag{7}$$

Using Lemma 2, we can get

$$f(O_2^+ \mid S_1) + f(O_2^- \mid S_1) + f(O_3 \mid S_1) + f(O_1^+ \mid S_2) + f(O_1^- \mid S_2) + f(O_4 \mid S_2)$$

$$\le \sum_{e \in O_1^+ \cup O_2^- \cup O_3} \delta(\pi_1(e)) + \sum_{e \in O_1^- \cup O_2^+ \cup O_4} \delta(\pi_2(e))$$

$$\le \sum_{e \in S_1} \delta(e) + \sum_{e \in S_2} \delta(e) \le f(S_1) + f(S_2), \tag{8}$$

where the second inequality is due to that $\pi_1 : O_1^+ \cup O_1^- \cup O_2^- \cup O_3 \mapsto S_1$ and $\pi_2 : O_1^- \cup O_2^+ \cup O_2^- \cup O_4 \mapsto S_2$ are both injective functions shown in Lemma 1, and that $\delta(e) > 0$ for every $e \in S_1 \cup S_2$ according to Line 8 of TwinGreedy. Besides, according to the definition of $O_5$, we must have $f(e \mid S_1) \le 0$ for each $e \in O_5$, because otherwise $e$ should be added into $S_1 \cup S_2$ as $S_1 \cup \{e\} \in \mathcal{I}$. Similarly, we get $f(e \mid S_2) \le 0$ for each $e \in O_6$. Therefore, we have

$$f(O_5 \mid S_1) \le \sum_{e \in O_5} f(e \mid S_1) \le 0; \;\; f(O_6 \mid S_2) \le \sum_{e \in O_6} f(e \mid S_2) \le 0 \tag{9}$$

Meanwhile, as $f(\cdot)$ is a non-negative submodular function and $S_1 \cap S_2 = \emptyset$, we have

$$f(O) \le f(O) + f(O \cup S_1 \cup S_2) \le f(O \cup S_1) + f(O \cup S_2) \tag{10}$$

**Algorithm 2:** TwinGreedyFast($\mathcal{N}, \mathcal{I}, f(\cdot), \epsilon$)

---

1   $\tau_{max} \leftarrow \max\{f(e) : e \in \mathcal{N} \wedge \{e\} \in \mathcal{I}\}$;

2   $S_1 \leftarrow \emptyset; S_2 \leftarrow \emptyset$;

3   **for** $(\tau \leftarrow \tau_{max}; \;\; \tau > \epsilon\tau_{max}/[r(1+\epsilon)]; \;\; \tau \leftarrow \tau/(1+\epsilon))$ **do**

4      **foreach** $e \in \mathcal{N} \backslash (S_1 \cup S_2)$ **do**

5          $\Delta_1 \leftarrow -\infty; \Delta_2 \leftarrow -\infty$     /*two signals*/

6          **if** $S_1 \cup \{e\} \in \mathcal{I}$ **then** $\Delta_1 \leftarrow f(e \mid S_1)$ ;

7          **if** $S_2 \cup \{e\} \in \mathcal{I}$ **then** $\Delta_2 \leftarrow f(e \mid S_2)$ ;

8          $i \leftarrow \arg\max_{j \in \{1,2\}} \Delta_j$; (ties broken arbitrarily)

9          **if** $\Delta_i \geq \tau$ **then** $S_i \leftarrow S_i \cup \{e\}$ ;

10   $S^* \leftarrow \arg\max_{X \in \{S_1, S_2\}} f(X)$

11   **return** $S^*$

---

By summing up Eqn. (6)-Eqn. (10) and simplifying, we get

$$f(O) \leq 2f(S_1) + 2f(S_2) \leq 4f(S^*) \tag{11}$$

which completes the proof on the $1/4$ ratio. Finally, the $\mathcal{O}(nr)$ time complexity is evident, as $|S_1| + |S_2| \leq 2r$ and adding one element into $S_1$ or $S_2$ has at most $\mathcal{O}(n)$ time complexity. $\qquad\square$

Interestingly, when the objective function $f(\cdot)$ is monotone, the proof of Theorem 1 shows that the TwinGreedy algorithm achieves a $1/2$ approximation ratio due to $f(O \cup S_1) + f(O \cup S_2) \geq 2f(O)$. Therefore, we rediscover the $1/2$ deterministic ratio proposed by Fisher et al. [27] for monotone submodular maximization over a matroid.

## 4   The TwinGreedyFast Algorithm

As the TwinGreedy algorithm still has quadratic running time, we present a more efficient algorithm TwinGreedyFast (Alg. 2). As that in TwinGreedy, the TwinGreedyFast algorithm also maintains two solution sets $S_1$ and $S_2$, but it uses a threshold to control the quality of elements added into $S_1$ or $S_2$. More specifically, given a threshold $\tau$, TwinGreedyFast checks every unselected element $e \in \mathcal{N} \backslash (S_1 \cup S_2)$ in an arbitrarily order. Then it chooses $S_i \in \{S_1, S_2\}$ such that adding $e$ into $S_i$ does not violate the feasibility of $\mathcal{I}$ while $f(e \mid S_i)$ is maximized. If the marginal gain $f(e \mid S_i)$ is no less than $\tau$, then $e$ is added into $S_i$, otherwise the algorithm simply neglects $e$. The algorithm repeats the above process starting from $\tau = \tau_{max}$ and decreases $\tau$ by a factor $(1 + \epsilon)$ at each iteration until $\tau$ is sufficiently small, then it returns the one between $S_1$ and $S_2$ with the larger objective value.

The performance analysis of TwinGreedyFast is similar to that of TwinGreedy. Let us consider the two solution sets $S_1$ and $S_2$ when TwinGreedyFast returns. We can define $O_1^+, O_1^-, O_2^+, O_2^-, O_3, O_4$, $\text{Pre}(e, S_i)$ and $\delta(e)$ in exact same way as Definition 1, and Lemma 1 still holds as the construction of $\pi_1(\cdot)$ and $\pi_2(\cdot)$ only depends on the insertion order of the elements in $S_1 \cup S_2$, which is fixed when TwinGreedyFast returns. We also try to bound the marginal gains of $O_1^+$-$O_4$ with respect to $S_1$ and $S_2$ in a way similar to Lemma 2. However, due to the thresholds introduced in TwinGreedyFast, these bounds are slightly different from those in Lemma 2, as shown in Lemma 3:

**Lemma 3** *The TwinGreedyFast algorithm satisfies:*

$$f(O_1^+ \mid S_2) \leq \sum_{e \in O_1^+} \delta(\pi_1(e)); \qquad f(O_2^+ \mid S_1) \leq \sum_{e \in O_2^+} \delta(\pi_2(e)) \tag{12}$$

$$f(O_1^- \mid S_2) \leq (1 + \epsilon) \sum_{e \in O_1^-} \delta(\pi_2(e)); \qquad f(O_2^- \mid S_1) \leq (1 + \epsilon) \sum_{e \in O_2^-} \delta(\pi_1(e)) \tag{13}$$

$$f(O_4 \mid S_2) \leq (1 + \epsilon) \sum_{e \in O_4} \delta(\pi_2(e)); \qquad f(O_3 \mid S_1) \leq (1 + \epsilon) \sum_{e \in O_3} \delta(\pi_1(e)) \tag{14}$$

*where $\pi_1(\cdot)$ and $\pi_2(\cdot)$ are the two functions defined in Lemma 1.*

**Proof**:   (sketch) At the moment that any $e \in O_1^+$ is added into $S_1$, it can also be added into $S_2$ due to Definition 1. Therefore, we must have $f(e \mid \text{Pre}(e, S_2)) \leq \delta(e)$ according to the greedy selection

rule of TwinGreedyFast. Using submodularity, we get the first inequality in the lemma:

$$f(O_1^+ \mid S_2) \leq \sum\nolimits_{e \in O_1^+} f(e \mid S_2) \leq \sum\nolimits_{e \in O_1^+} f(e \mid \mathrm{Pre}(e, S_2)) \leq \sum\nolimits_{e \in O_1^+} \delta(\pi_1(e))$$

For any $e \in O_1^-$, consider the moment that TwinGreedyFast adds $\pi_2(e)$ into $S_2$. According to Lemma 1, we have $\mathrm{Pre}(\pi_2(e), S_2) \cup \{e\} \in \mathcal{I}$. This implies that $e$ has not been added into $S_1$ yet, because otherwise we have $\mathrm{Pre}(e, S_2) \subseteq \mathrm{Pre}(\pi_2(e), S_2)$ and hence $\mathrm{Pre}(e, S_2) \cup \{e\} \in \mathcal{I}$ according to the hereditary property of independence systems, which contradicts $e \in O_1^-$. As such, we must have $\delta(\pi_2(e)) \geq \tau$ and $f(e \mid \mathrm{Pre}(\pi_2(e), S_2)) \leq (1 + \epsilon)\tau$ where $\tau$ is the threshold used by TwinGreedyFast when adding $\pi_2(e)$, because otherwise $e$ should have been added before $\pi_2(e)$ in an earlier stage of the algorithm with a larger threshold. Combining these results gives us

$$f(O_1^- \mid S_2) \leq \sum\nolimits_{e \in O_1^-} f(e \mid S_2) \leq \sum\nolimits_{e \in O_1^-} f(e \mid \mathrm{Pre}(\pi_2(e), S_2)) \leq (1 + \epsilon) \sum\nolimits_{e \in O_1^-} \delta(\pi_2(e))$$

The other inequalities in the lemma can be proved similarly. $\qquad\square$

With Lemma 3, we can use similar reasoning as that in Theorem 1 to prove the performance bounds of TwinGreedyFast, as shown in Theorem 2. The full proofs of Lemma 3 and Theorem 2 can be found in the supplementary file.

**Theorem 2** *When $(\mathcal{N}, \mathcal{I})$ is a matroid, the* TwinGreedyFast *algorithm returns a solution $S^*$ with $\frac{1}{4} - \epsilon$ approximation ratio, under time complexity of $\mathcal{O}(\frac{n}{\epsilon} \log \frac{r}{\epsilon})$.*

**Extensions:** The TwinGreedyFast algorithm can also be directly used to address the problem of non-monotone submodular maximization subject to a $p$-set system constraint (by simply inputting a $p$-set system $(\mathcal{N}, \mathcal{I})$ into TwinGreedyFast). It achieves a $\frac{1}{2p+2} - \epsilon$ deterministic ratio under $\mathcal{O}(\frac{n}{\epsilon} \log \frac{r}{\epsilon})$ time complexity in such a case, which improves upon both the ratio and time complexity of the results in [31, 44]. We note that the prior best known result for this problem is a $\left(\frac{1}{p+2\sqrt{p}+3} - \epsilon\right)$-approximation under $\mathcal{O}((nr + r/\epsilon)\sqrt{p})$ time complexity proposed in [25]. Compared to this best known result, TwinGreedyFast achieves much smaller time complexity, and also has a better approximation ratio when $p \leq 5$. The proof for this performance ratio of TwinGreedyFast (shown in the supplementary file) is almost the same as those for Theorems 1-2, as we only need to relax Lemma 1 to allow that the preimage by $\pi_1(\cdot)$ or $\pi_2(\cdot)$ of any element in $S_1 \cup S_2$ contains at most $p$ elements.

## 5 Performance Evaluation

In this section, we evaluate the performance of TwinGreedyFast under two social network applications, using both synthetic and real networks. We implement the following algorithms for comparison:

1. **SampleGreedy**: This algorithm is proposed in [25] which has $1/4$ expected ratio and $\mathcal{O}(nr)$ running time. To the best of our knowledge, it is currently the fastest randomized algorithm for non-monotone submodular maximization over a matroid.

2. **Fantom**: Excluding the $1/4 - \epsilon$ ratio proposed in [38], the Fantom algorithm from [44] has the best deterministic ratio for our problem. As it needs to call an unconstrained submodular maximization (USM) algorithm, we use the USM algorithm proposed in [12] with $1/3$ deterministic ratio and linear running time. As such, Fantom achieves $1/7$ deterministic ratio in the experiments.[2]

3. **ResidualRandomGreedy**: This algorithm is from [11] which has $1/4$ expected ratio and $\mathcal{O}(nr \log n)$ running time. We denote it as "RRG" for brevity.

4. **TwinGreedyFast**: We implement our Algorithm 2 by setting $\epsilon = 0.1$. As such, it achieves $0.15$ deterministic approximation ratio in the experiments.

## 5.1 Applications

Given a social network $G = (V, E)$ where $V$ is the set of nodes and $E$ is the set of edges, we consider the following two applications in our experiments. Both applications are instances of non-monotone submodular maximization subject to a matroid constraint, which is proved in the supplementary file.

1. Social Network Monitoring: This application is similar to those applications considered in [40] and [36]. Suppose that each edge $(u, v) \in E$ is associated with a weight $w(u, v)$ denoting the maximum amount of content that can be propagated through $(u, v)$. Moreover, $V$ is partitioned into $h$ disjoint subsets $V_1, V_2, \cdots, V_h$ according to the users' properties such as ages and political leanings. We need to select a set of users $S \subseteq V$ to monitor the network, such that the total amount of monitored content $f(S) = \sum_{(u,v) \in E, u \in S, v \notin S} w(u, v)$ is maximized. Due to the considerations on diversity or fairness, we also require $\forall i \in [h] : |S \cap V_i| \leq k$, where $k$ is a predefined constant.

2. Multi-Product Viral Marketing: This application is a variation of the one considered in [18]. Suppose that a company with a budget $B$ needs to select a set of at most $k$ seed nodes to promote $m$ products. Each node $u \in V$ can be selected as a seed for at most one product and also has a cost $c(u)$ for serving as a seed. The goal is to maximize the revenue (with budget-saving considerations): $\sum_{i \in [m]} f_i(S_i) + \left( B - \sum_{i \in [m]} \sum_{v \in S_i} c(v) \right)$, where $S_i$ is the set of seed nodes selected for product $i$, and $f_i(\cdot)$ is a monotone and submodular influence spread function as that proposed in [34]. We also assume that $B$ is large enough to keep the revenue non-negative, and assume that selecting no seed nodes would cause zero revenue.

## 5.2 Experimental Results

The experimental results are shown in Fig. 2. In overview, TwinGreedyFast runs more than an order of magnitude faster than the other three algorithms; Fantom has the best performance on utility; the performance of TwinGreedyFast on utility is close to that of Fantom.

### 5.2.1 Social Network Monitoring

We generate an Erdős–Rényi (ER) random graph with 3000 nodes and set the edge probability to 0.5. The weight $w(u, v)$ for each $(u, v) \in E$ is generated uniformly at random from $[0, 1]$ and all nodes are randomly assigned to five groups. In the supplementary file, we also generate Barabási–Albert (BA) graphs for comparison, and the results are qualitatively similar. It can be seen from Fig. 2(a) that Fantom incurs the largest number of queries to objective function, as it leverages repeated greedy searching processes to find a solution with good quality. SampleGreedy runs faster than RRG, which coincides with their time complexity mentioned in Section 1.1. TwinGreedyFast significantly outperforms Fantom, RRG and SampleGreedy by more than an order of magnitude in Fig. 2(a) and Fig. 2(b), as it achieves nearly linear time complexity. Moreover, it can be seen from Figs. 2(a)-(b) that TwinGreedyFast maintains its advantage on efficiency no matter the metric is wall-clock running time or number of queries. Finally, Fig. 2(c) shows that all the implemented algorithms achieve approximately the same utility on the ER random graph.

### 5.2.2 Multi-Product Viral Marketing

We use two real social networks Flixster and Epinions. Flixster is from [6] with 137925 nodes and 2538746 edges, and Epinions is from the SNAP dataset collection [39] with 75879 nodes and 508837 edges. We consider three products and follow the Independent Cascade (IC) model [34] to set the influence spread function $f_i(\cdot)$ for each product $i$. Note that the IC model requires an activation probability $p_{u,v}$ associated with each edge $(u, v) \in E$. This probability is available in the Flixster dataset (learned from real users' action logs), and we follow Chen et al. [17] to set $p_{u,v} = 1/|N_{in}(v)|$ for the Epinions dataset, where $N_{in}(v)$ is the set of in-neighbors of $v$. As evaluating $f_i(\cdot)$ under the IC model is NP-hard, we follow the approach in [7] to generate a set of one million random Reverse-Reachable Sets (RR-sets) such that $f_i(\cdot)$ can be approximately evaluated using these RR-sets. The cost $c(u)$ of each node is generated uniformly at random from $[0, 1]$, and we set $B = m \sum_{u \in V} c(u)$ to keep the function value non-negative. More implementation details can be found in the supplementary file.

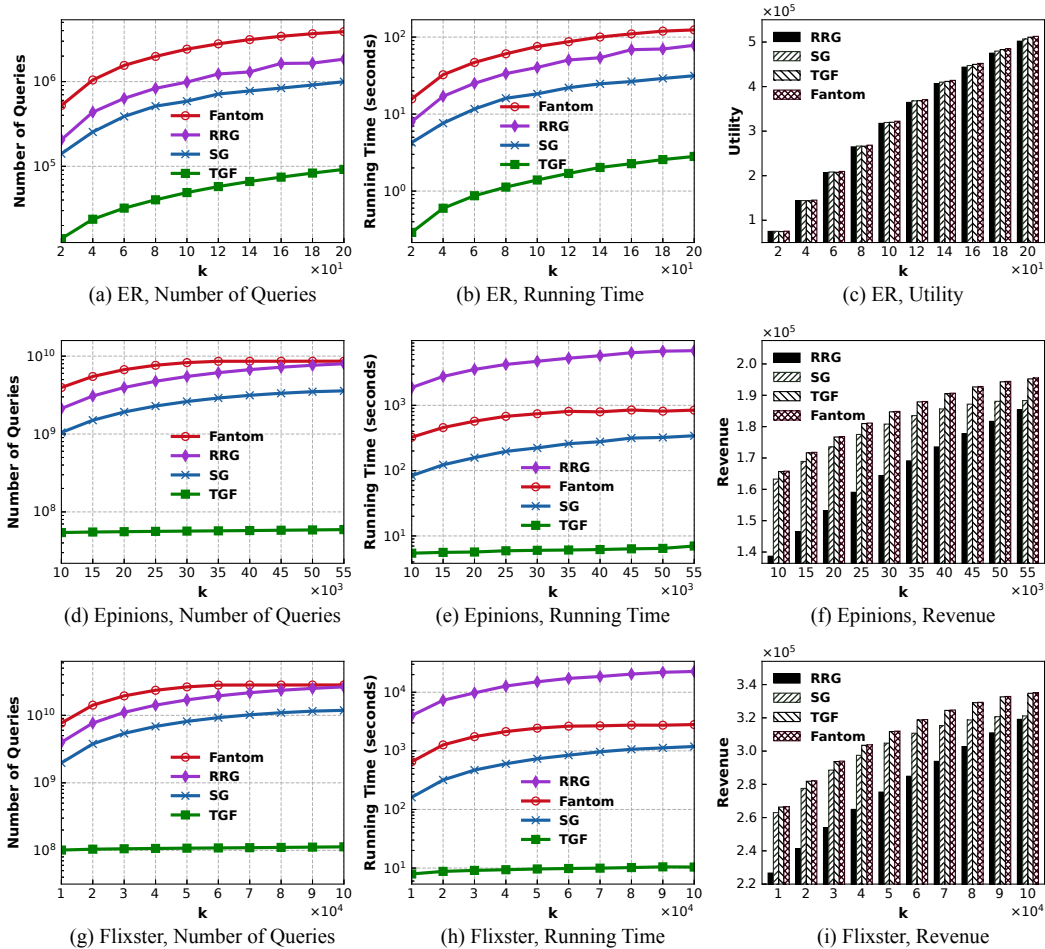

Figure 2: Experimental results for social network monitoring and multi-product viral marketing, where "TGF" and "SG" are abbreviations for "TwinGreedyFast" and "SampleGreedy", respectively.

We plot the experimental results on viral marketing in Fig. 2(d)-Fig. 2(i). The results are qualitatively similar to those in Fig. 2(a)-Fig. 2(c), which show that TwinGreedyFast is still significantly faster than the other algorithms. Fantom achieves the largest utility again, while TwinGreedyFast performs closely to Fantom and outperforms the other two randomized algorithms on utility.

## 6 Conclusion and Discussion

We have proposed the first deterministic algorithm to achieve an approximation ratio of $1/4$ for maximizing a non-monotone, non-negative submodular function subject to a matroid constraint, and our algorithm can also be accelerated to achieve nearly-linear running time. In contrast to the existing algorithms adopting the "repeated greedy-search" framework proposed by [31], our algorithms are designed based on a novel "simultaneous greedy-search" framework, where two candidate solutions are constructed simultaneously, and a pair of an element and a candidate solution is greedily selected at each step to maximize the marginal gain. Moreover, our algoirthms can also be directly used to handle a more general $p$-set system constraint or monotone submodular functions, while still achieving nice performance bounds. For example, by similar reasoning with that in Sections 3-4, it can be seen that our algorithms can achieve $\frac{1}{p+1}$ ratio for monotone $f(\cdot)$ under a $p$-set system constraint, which is almost best possible [2]. We have evaluated the performance of our algorithms in two concrete applications for social network monitoring and multi-product viral marketing, and the extensive experimental results demonstrate that our algorithms runs in orders of magnitude faster than the state-of-the-art algorithms, while achieving approximately the same utility.

## Acknowledgements

This work was supported by the National Key R&D Program of China under Grant No. 2018AAA0101204, the National Natural Science Foundation of China (NSFC) under Grant No. 61772491 and Grant No. U1709217, Anhui Initiative in Quantum Information Technologies under Grant No. AHY150300, and the Fundamental Research Funds for the Central Universities.

## Broader Impact

Submodular optimization is an important research topic in data mining, machine learning and optimization theory, as it has numerous applications such as crowdsourcing [47], viral marketing [34], feature selection [28], network monitoring [40], document summarization [20, 41], online advertising [49], crowd teaching [46] and blogosphere mining [21]. Matroid is a fundamental structure in combinatorics that captures the essence of a notion of "independence" that generalizes linear independence in vector spaces. The matroid structure has been pervasively found in various areas such as geometry, network theory, coding theory and graph theory [8]. The study on submodular maximization subject to matroid constraints dates back to the 1970s (e.g., [27]), and it is still a hot research topic today [3, 5, 16, 25, 44, 45]. A lot of practical problems can be cast as the problem of submodular maximziation over a matroid constraint (or more general $p$-set system constraints), such as diversity maximization [1], video summarization [52], clustering [42], multi-robot allocation [51] and planning sensor networks [19]. Therefore, our study has addressed a general and fundamental theoretical problem with many potential applications.

Due to the massive datasets used everywhere nowadays, it is very important that submodular optimization algorithms should achieve accuracy and efficiency simultaneously. Recently, there emerge great interests on designing more practical and efficient algorithms for submodular optimization (e.g., [2, 4, 11, 13, 23, 36, 43]), and our work advances the state of the art in this area by proposing a new efficient algorithm with improved performance bounds. Moreover, our algorithms are based on a novel "simultaneous greedy-search" framework, which is different from the classical "repeated greedy-search" and "local search" frameworks adopted by the state-of-the-art algorithms (e.g., [25, 31, 38, 44]). We believe that our "simultaneous greedy-search" framework has the potential to be extended to address other problems on submodular maximization with more complex constraints, which is the topic of our ongoing research.

## Footnotes

[1]Although the randomized 0.283-approximation algorithm in [11] has a better approximation ratio than TwinGreedy, its time complexity is larger than that of TwinGreedy by at least an additive factor of $\mathcal{O}(r^3)$, which can be large as $r$ can be in the order of $\Theta(n)$.

[2]We have also tested Fantom using the randomized USM algorithm in [12] with $1/2$ expected ratio. The experimental results are almost identical, although Fantom has a larger $1/6$ ratio (in expectation) in such a case.

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
