[Supplementary Material]

## Appendix A: Missing Proofs

### A.1 Proof of Lemma 1

***Proof***: We only prove the existence of $\pi_1(\cdot)$, as the existence of $\pi_2(\cdot)$ can be proved in the same way. Suppose that the elements in $S_1$ are $\{u_1, \cdots, u_s\}$ (listed according to the order that they are added into $S_1$). We use an argument inspired by [15] to construct $\pi_1(\cdot)$. Let $L_s = O_1^+ \cup O_1^- \cup O_2^- \cup O_3$. We execute the following iterations from $j = s$ to $j = 0$. At the beginning of the $j$-th iteration, we compute a set $A_j = \{x \in L_j \backslash \{u_1, \cdots, u_{j-1}\} : \{u_1, \cdots, u_{j-1}, x\} \in \mathcal{I}\}$. If $u_j \in O_1^+ \cup O_1^-$ (so $u_j \in A_j$), then we set $\pi_1(u_j) = u_j$ and $D_j = \{u_j\}$. If $u_j \notin O_1^+ \cup O_1^-$ and $A_j \neq \emptyset$, then we pick an arbitrary $e \in A_j$ and set $\pi_1(e) = u_j$; $D_j = \{e\}$. If $A_j = \emptyset$, then we simply set $D_j = \emptyset$. After that, we set $L_{j-1} = L_j \backslash D_j$ and enter the $(j-1)$-th iteration.

From the above process, it can be easily seen that $\pi_1(\cdot)$ has the properties required by the lemma as long as it is a valid function. So we only need to prove that each $e \in O_1^+ \cup O_1^- \cup O_2^- \cup O_3$ is mapped to an element in $S_1$, which is equivalent to prove $L_0 = \emptyset$ as each $e \in L_s \backslash L_0$ is mapped to an element in $S_1$ according to the above process. In the following, we prove $L_0 = \emptyset$ by induction, i.e., proving $|L_j| \leq j$ for all $0 \leq j \leq s$.

We first prove $|L_s| \leq s$. By way of contradiction, let us assume $|L_s| = |O_1^+ \cup O_1^- \cup O_2^- \cup O_3| > s = |S_1|$. Then, there must exist some $x \in O_2^- \cup O_3$ satisfying $S_1 \cup \{x\} \in \mathcal{I}$ according to the exchange property of matroids. Moreover, according to the definition of $O_3$, we also have $x \notin O_3$, which implies $x \in O_2^-$. So we can get $\mathrm{Pre}(x, S_1) \cup \{x\} \in \mathcal{I}$ due to $\mathrm{Pre}(x, S_1) \subseteq S_1$, $S_1 \cup \{x\} \in \mathcal{I}$ and the hereditary property of independence systems, but this contradicts the definition of $O_2^-$. Therefore, $|L_j| \leq j$ holds when $j = s$.

Now suppose $|L_j| \leq j$ for certain $j \leq s$. If $A_j \neq \emptyset$, then we have $D_j \neq \emptyset$ and hence $|L_{j-1}| = |L_j| - 1 \leq j - 1$. If $A_j = \emptyset$, then we know that there does not exist $x \in L_j \backslash \{u_1, \cdots, u_{j-1}\}$ such that $\{u_1, \cdots, u_{j-1}\} \cup \{x\} \in \mathcal{I}$. This implies $|\{u_1, \cdots, u_{j-1}\}| \geq |L_j|$ due to the exchange property of matroids. So we also have $|L_{j-1}| = |L_j| \leq j - 1$, which completes the proof. $\square$

### A.2 Proof of Lemma 2

For clarity, we decompose Lemma 2 into three lemmas (Lemmas 4-6) and prove each of them.

**Lemma 4** *The TwinGreedy algorithm satisfies*

$$f(O_1^+ \mid S_2) \leq \sum_{e \in O_1^+} \delta(\pi_1(e)); \quad f(O_2^+ \mid S_1) \leq \sum_{e \in O_2^+} \delta(\pi_2(e)) \tag{15}$$

***Proof of Lemma 4***: We only prove the first inequality, as the second one can be proved in the same way. For any $e \in O_1^+$, consider the moment that TwinGreedy inserts $e$ into $S_1$. At that moment, adding $e$ into $S_2$ also does not violate the feasibility of $\mathcal{I}$ according to the definition of $O_1^+$. Therefore, we must have $f(e \mid \mathrm{Pre}(e, S_2)) \leq \delta(e)$, because otherwise $e$ would not be inserted into $S_1$ according to the greedy rule of TwinGreedy. Using submodularity and the fact that $\pi_1(e) = e$, we get

$$f(e \mid S_2) \leq f(e \mid \mathrm{Pre}(e, S_2)) \leq \delta(e) = \delta(\pi_1(e)), \quad \forall e \in O_1^+ \tag{16}$$

and hence

$$\sum_{e \in O_1^+} f(O_1^+ \mid S_2) \leq \sum_{e \in O_1^+} f(e \mid S_2) \leq \sum_{e \in O_1^+} \delta(\pi_1(e)), \tag{17}$$

which completes the proof. $\square$

**Lemma 5** *The TwinGreedy algorithm satisfies*

$$f(O_1^- \mid S_2) \leq \sum_{e \in O_1^-} \delta(\pi_2(e)); \quad f(O_2^- \mid S_1) \leq \sum_{e \in O_2^-} \delta(\pi_1(e)) \tag{18}$$

***Proof of Lemma 5***:     We only prove the first inequality, as the second one can be proved in the same way. For any $e \in O_1^-$, consider the moment that TwinGreedy inserts $\pi_2(e)$ into $S_2$. At that moment, adding $e$ into $S_2$ also does not violate the feasibility of $\mathcal{I}$ as $\mathrm{Pre}(\pi_2(e), S_2) \cup \{e\} \in \mathcal{I}$ according to Lemma 1. This implies that $e$ has not been inserted into $S_1$ yet. To see this, let us assume (by way of contradiction) that $e$ has already been added into $S_1$ when TwinGreedy inserts $\pi_2(e)$ into $S_2$. So we have $\mathrm{Pre}(e, S_2) \subseteq \mathrm{Pre}(\pi_2(e), S_2)$. As $\mathrm{Pre}(\pi_2(e), S_2) \cup \{e\} \in \mathcal{I}$, we must have $\mathrm{Pre}(e, S_2) \cup \{e\} \in \mathcal{I}$ due to the hereditary property of independence systems. However, this contradicts $\mathrm{Pre}(e, S_2) \cup \{e\} \notin \mathcal{I}$ as $e \in O_1^-$.

As $e$ has not been inserted into $S_1$ yet at the moment that $\pi_2(e)$ is inserted into $S_2$, and $\mathrm{Pre}(\pi_2(e), S_2) \cup \{e\} \in \mathcal{I}$, we know that $e$ must be a contender to $\pi_2(e)$ when TwinGreedy inserts $\pi_2(e)$ into $S_2$. Due to the greedy selection rule of the algorithm, this means $\delta(\pi_2(e)) = f(\pi_2(e) \mid \mathrm{Pre}(\pi_2(e), S_2)) \geq f(e \mid \mathrm{Pre}(\pi_2(e), S_2))$. As $\mathrm{Pre}(\pi_2(e), S_2) \subseteq S_2$, we also have $f(e \mid \mathrm{Pre}(\pi_2(e), S_2)) \geq f(e \mid S_2)$. Putting these together, we have

$$f(O_1^- \mid S_2) \leq \sum_{e \in O_1^-} f(e \mid S_2) \leq \sum_{e \in O_1^-} f(e \mid \mathrm{Pre}(\pi_2(e), S_2)) \leq \sum_{e \in O_1^-} \delta(\pi_2(e)) \qquad (19)$$

which completes the proof. $\qquad\square$

**Lemma 6** *The TwinGreedy algorithm satisfies*

$$f(O_3 \mid S_1) \leq \sum_{e \in O_3} \delta(\pi_1(e)); \;\; f(O_4 \mid S_2) \leq \sum_{e \in O_4} \delta(\pi_2(e)) \qquad (20)$$

***Proof of Lemma 6***:     We only prove the first inequality, as the second one can be proved in the same way. Consider any $e \in O_3$. According to Lemma 1, we have $\mathrm{Pre}(\pi_1(e), S_1) \cup \{e\} \in \mathcal{I}$, which means that $e$ can be added into $S_1$ without violating the feasibility of $\mathcal{I}$ when $\pi_1(e)$ is added into $S_1$. According to the greedy rule of TwinGreedy and submodularity, we must have $\delta(\pi_1(e)) = f(\pi_1(e) \mid \mathrm{Pre}(\pi_1(e), S_1)) \geq f(e \mid \mathrm{Pre}(\pi_1(e), S_1))$, because otherwise $e$ should be added into $S_1$, which contradicts $e \in O_3$. Therefore, we get

$$f(O_3 \mid S_1) \leq \sum_{e \in O_3} f(e \mid S_1) \leq \sum_{e \in O_3} f(e \mid \mathrm{Pre}(\pi_1(e), S_1)) \leq \sum_{e \in O_3} \delta(\pi_1(e)) \qquad (21)$$

which completes the proof. $\qquad\square$

## A.3   Proof of Theorem 1

***Proof***:     We only consider the special case that $S_1$ or $S_2$ is empty, as the main proof of the theorem has been presented in the paper. Without loss of generality, we assume that $S_2$ is empty. According to the greedy rule of the algorithm, we have

$$f(O \cap S_1 \mid \emptyset) \leq \sum_{e \in O \cap S_1} f(e \mid \emptyset) \leq \sum_{e \in O \cap S_1} \delta(e) \leq \sum_{e \in S_1} \delta(e) = f(S_1 \mid \emptyset) \qquad (22)$$

and $f(O \backslash S_1 \mid \emptyset) \leq \sum_{e \in O \backslash S_1} f(e \mid \emptyset) \leq 0$. Combining these with

$$f(O \backslash S_1) + f(O \cap S_1) \geq f(O) + f(\emptyset), \qquad (23)$$

we get $f(S_1) \geq f(O)$, which proves that $S_1$ is an optimal solution when $S_2$ is empty. $\qquad\square$

## A.4   Proof of Lemma 3

For clarity, we decompose Lemma 3 into three lemmas (Lemmas 7-9) and prove each of them.

**Lemma 7** *For the TwinGreedyFast algorithm, we have*

$$f(O_1^+ \mid S_2) \leq \sum_{e \in O_1^+} \delta(\pi_1(e)); \;\; f(O_2^+ \mid S_1) \leq \sum_{e \in O_2^+} \delta(\pi_2(e)) \qquad (24)$$

***Proof of Lemma 7***:    The proof is similar to that of Lemma 4, and we present the full proof for completeness. We will only prove the first inequality, as the second one can be proved in the same way. For any $e \in O_1^+$, consider the moment that TwinGreedy inserts $e$ into $S_1$ and suppose that the current threshold is $\tau$. Therefore, we must have $\delta(e) \geq \tau$. At that moment, adding $e$ into $S_2$ also does not violate the feasibility of $\mathcal{I}$ according to the definition of $O_1^+$. So we must have $f(e \mid \mathrm{Pre}(e, S_2)) \leq \delta(e)$, because otherwise we have $f(e \mid \mathrm{Pre}(e, S_2)) > \delta(e) \geq \tau$, and hence $e$ would be inserted into $S_2$ according to the greedy rule of TwinGreedyFast. Using submodularity and the fact that $\pi_1(e) = e$, we get

$$f(e \mid S_2) \leq f(e \mid \mathrm{Pre}(e, S_2)) \leq \delta(e) = \delta(\pi_1(e)), \quad \forall e \in O_1^+ \tag{25}$$

and hence

$$\sum_{e \in O_1^+} f(O_1^+ \mid S_2) \leq \sum_{e \in O_1^+} f(e \mid S_2) \leq \sum_{e \in O_1^+} \delta(\pi_1(e)), \tag{26}$$

which completes the proof.                                                        □

**Lemma 8** *For the TwinGreedyFast Algorithm, we have*

$$f(O_1^- \mid S_2) \leq (1 + \epsilon) \sum_{e \in O_1^-} \delta(\pi_2(e)); \ \ f(O_2^- \mid S_1) \leq (1 + \epsilon) \sum_{e \in O_2^-} \delta(\pi_1(e)) \tag{27}$$

***Proof of Lemma 8***:    We only prove the first inequality, as the second one can be proved in the same way. For any $e \in O_1^-$, consider the moment that TwinGreedyFast adds $\pi_2(e)$ into $S_2$. Using the same reasoning with that in Lemma 5, we can prove: (1) $e$ has not been inserted into $S_1$ at the moment that $\pi_2(e)$ is inserted into $S_2$; (2) $\mathrm{Pre}(\pi_2(e), S_2) \cup \{e\} \in \mathcal{I}$ (due to Lemma 1).

Let $\tau$ be the threshold set by the algorithm when $\pi_2(e)$ is inserted into $S_2$. So we must have $\delta(\pi_2(e)) \geq \tau$. Moreover, we must have $f(e \mid \mathrm{Pre}(\pi_2(e), S_2)) \leq (1 + \epsilon)\tau$. To see this, let us assume $f(e \mid \mathrm{Pre}(\pi_2(e), S_2)) > (1 + \epsilon)\tau$ by way of contradiction. If $\tau = \tau_{max}$, then we get $f(e) \geq f(e \mid \mathrm{Pre}(\pi_2(e), S_2)) > (1 + \epsilon)\tau_{max}$, which contradicts $f(e) \leq \tau_{max}$. If $\tau < \tau_{max}$, then consider the moment that $e$ is checked by the TwinGreedyFast algorithm when the threshold is $\tau' = (1 + \epsilon)\tau$. Let $S_{2,\tau'}$ be the set of elements in $S_2$ at that moment. Then we have $f(e \mid S_{2,\tau'}) \geq \tau'$ due to $S_{2,\tau'} \subseteq \mathrm{Pre}(\pi_2(e), S_2)$ and submodularity of $f(\cdot)$. Moreover, we must have $S_{2,\tau'} \cup \{e\} \in \mathcal{I}$ due to $\mathrm{Pre}(\pi_2(e), S_2) \cup \{e\} \in \mathcal{I}$ and the hereditary property of independence systems. Consequently, $e$ should have be added by the algorithm when the threshold is $\tau'$, which contradicts the fact stated above that $e$ has not been added into $S_1$ at the moment that $\pi_2(e)$ is inserted into $S_2$ (under the threshold $\tau$). According to the above reasoning, we get

$$f(O_1^- \mid S_2) \leq \sum_{e \in O_1^-} f(e \mid S_2) \leq \sum_{e \in O_1^-} f(e \mid \mathrm{Pre}(\pi_2(e), S_2)) \leq (1 + \epsilon) \sum_{e \in O_1^-} \delta(\pi_2(e)) \tag{28}$$

which completes the proof.                                                        □

**Lemma 9** *For the TwinGreedyFast algorithm, we have*

$$f(O_3 \mid S_1) \leq (1 + \epsilon) \sum_{e \in O_3} \delta(\pi_1(e)); \ \ f(O_4 \mid S_2) \leq (1 + \epsilon) \sum_{e \in O_4} \delta(\pi_2(e)) \tag{29}$$

***Proof of Lemma 9***:    We only prove the first inequality, as the second one can be proved in the same way. Consider any $e \in O_3$. According to Lemma 1, we have $\mathrm{Pre}(\pi_1(e), S_1) \cup \{e\} \in \mathcal{I}$, i.e., $e$ can be added into $S_1$ without violating the feasibility of $\mathcal{I}$ when $\pi_1(e)$ is added into $S_1$. By similar reasoning with that in Lemma 8, we can get $f(e \mid \mathrm{Pre}(\pi_1(e), S_1)) \leq (1 + \epsilon)\delta(\pi_1(e))$, because otherwise $e$ must have been added into $S_1$ in an earlier stage of the TwinGreedyFast algorithm (under a larger threshold) before $\pi_1(e)$ is added into $S_1$, but this contradicts $e \notin S_1 \cup S_2$. Therefore, we get

$$f(O_3 \mid S_1) \leq \sum_{e \in O_3} f(e \mid S_1) \leq \sum_{e \in O_3} f(e \mid \mathrm{Pre}(\pi_1(e), S_1)) \leq (1 + \epsilon) \sum_{e \in O_3} \delta(\pi_1(e)), \tag{30}$$

which completes the proof.                                                        □

## A.5 Proof of Theorem 2

***Proof***: In Theorem 3 of Appendix B, we will prove the performance bounds of TwinGreedyFast under a $p$-set system constraint. The proof of Theorem 3 can also be used to prove Theorem 2, simply by setting $p = 1$. □

## Appendix B: Extensions for a $p$-Set System Constraint

When the independence system $(\mathcal{N}, \mathcal{I})$ input to the TwinGreedyFast algorithm is a $p$-set system, it returns a solution $S^*$ achieving $\frac{1}{2p+2} - \epsilon$ approximation ratio. To prove this, we can define $O_1^+$, $O_1^-$, $O_2^+$, $O_2^-$, $O_3$, $O_4$, $\mathrm{Pre}(e, S_i)$ and $\delta(e)$ in exact same way as Definition 1, and then propose Lemma 10, which relaxes Lemma 1 to allow that the preimage by $\pi_1(\cdot)$ or $\pi_2(\cdot)$ of any element in $S_1 \cup S_2$ contains at most $p$ elements. The proof of Lemma 10 is similar to that of Lemma 1. For the sake of completeness and clarity, We provide the full proof of Lemma 10 in the following:

**Lemma 10** *There exist a function $\pi_1 : O_1^+ \cup O_1^- \cup O_2^- \cup O_3 \mapsto S_1$ such that:*

1. *For any $e \in O_1^+ \cup O_1^- \cup O_2^- \cup O_3$, we have $\mathrm{Pre}(\pi_1(e), S_1) \cup \{e\} \in \mathcal{I}$.*

2. *For each $e \in O_1^+ \cup O_1^-$, we have $\pi_1(e) = e$.*

3. *Let $\pi_1^{-1}(y) = \{e \in O_1^+ \cup O_1^- \cup O_2^- \cup O_3 : \pi_1(e) = y\}$ for any $y \in S_1$. Then we have $|\pi_1^{-1}(y)| \leq p$ for any $y \in S_1$.*

*Similarly, there exists a function $\pi_2 : O_1^- \cup O_2^+ \cup O_2^- \cup O_4 \mapsto S_2$ such that $\mathrm{Pre}(\pi_2(e), S_2) \cup \{e\} \in \mathcal{I}$ for each $e \in O_1^- \cup O_2^+ \cup O_2^- \cup O_4$ and $\pi_2(e) = e$ for each $e \in O_2^+ \cup O_2^-$ and $|\pi_2^{-1}(y)| \leq p$ for each $y \in S_2$.*

***Proof of Lemma 10***: We only prove the existence of $\pi_1(\cdot)$, as the existence of $\pi_2(\cdot)$ can be proved in the same way. Suppose that the elements in $S_1$ are $\{u_1, \cdots, u_s\}$ (listed according to the order that they are added into $S_1$). We use an argument inspired by [15] to construct $\pi_1(\cdot)$. Let $L_s = O_1^+ \cup O_1^- \cup O_2^- \cup O_3$. We execute the following iterations from $j = s$ to $j = 0$. At the beginning of the $j$-th iteration, we compute a set $A_j = \{x \in L_j \backslash \{u_1, \cdots, u_{j-1}\} : \{u_1, \cdots, u_{j-1}, x\} \in \mathcal{I}\}$. If $|A_j| \leq p$, then we set set $D_j = A_j$; if $|A_j| > p$ and $u_j \in O_1^+ \cup O_1^-$ (so $u_j \in A_j$), then we pick a subset $D_j \subseteq A_j$ satisfying $|D_j| = p$ and $u_j \in D_j$; if $|A_j| > p$ and $u_j \notin O_1^+ \cup O_1^-$, then we pick a subset $D_j \subseteq A_j$ satisfying $|D_j| = p$. After that, we set $\pi_1(e) = u_j$ for each $e \in D_j$ and set $L_{j-1} = L_j \backslash D_j$, and then enter the $(j-1)$-th iteration.

From the above process, it can be easily seen that Condition 1-3 in the lemma are satisfied. So we only need to prove that each $e \in O_1^+ \cup O_1^- \cup O_2^- \cup O_3$ is mapped to an element in $S_1$, which is equivalent to prove $L_0 = \emptyset$ as each $e \in L_s \backslash L_0$ is mapped to an element in $S_1$ according to the above process. In the following, we will prove $L_0 = \emptyset$ by induction, i.e., proving $|L_j| \leq pj$ for all $0 \leq j \leq s$.

When $j = s$, consider the set $M = S_1 \cup O_2^- \cup O_3$. Clearly, each element $e \in O_3$ satisfies $S_1 \cup \{e\} \notin \mathcal{I}$ according to the definition of $O_3$. Besides, we must have $S_1 \cup \{x\} \notin \mathcal{I}$ for each $x \in O_2^-$, because otherwise there exists $e \in O_2^-$ satisfying $S_1 \cup \{e\} \in \mathcal{I}$, and hence we get $\mathrm{Pre}(e, S_1) \cup \{e\} \in \mathcal{I}$ due to $\mathrm{Pre}(e, S_1) \subseteq S_1$ and the hereditary property of independence systems; contradicting $e \in O_2^-$. Therefore, we know that $S_1$ is a base of $M$. As $O_1^+ \cup O_1^- \cup O_2^- \cup O_3 \in \mathcal{I}$ and $O_1^+ \cup O_1^- \cup O_2^- \cup O_3 \subseteq M$, we get $|L_s| = |O_1^+ \cup O_1^- \cup O_2^- \cup O_3| \leq p|S_1| = ps$ according to the definition of $p$-set system.

Now suppose that $|L_j| \leq pj$ for certain $j \leq s$. If $|A_j| > p$, then we have $|D_j| = p$ and hence $|L_{j-1}| = |L_j| - p \leq p(j-1)$. If $|A_j| \leq p$, then we know that there does not exist $x \in L_{j-1} \backslash \{u_1, \cdots, u_{j-1}\}$ such that $\{u_1, \cdots, u_{j-1}\} \cup \{x\} \in \mathcal{I}$ due to the above process for constructing $\pi_1(\cdot)$. Now consider the set $M' = \{u_1, \cdots, u_{j-1}\} \cup L_{j-1}$, we know that $\{u_1, \cdots, u_{j-1}\}$ is a base of $M'$ and $L_{j-1} \in \mathcal{I}$, which implies $|L_{j-1}| \leq p(j-1)$ according to the definition of $p$-set system.

The above reasoning proves $|L_j| \leq pj$ for all $0 \leq j \leq s$ by induction, so we get $L_0 = \emptyset$ and hence the lemma follows. $\qquad\square$

With Lemma 10, Lemma 3 still holds under a $p$-set system constraint, as the proof of Lemma 3 only uses the hereditary property of independence systems and does not require that the functions $\pi_1(\cdot)$ and $\pi_2(\cdot)$ are injective. Therefore, we can still use Lemma 3 to prove the performance bounds of TwinGreedyFast under a $p$-set system constraint, as shown in Theorem 3. Note that the proof of Theorem 3 can also be used to prove Theorem 2, simply by setting $p = 1$.

**Theorem 3** *When the independence system $(\mathcal{N}, \mathcal{I})$ input to TwinGreedyFast is a $p$-set system, the TwinGreedyFast algorithm returns a solution $S^*$ with $\frac{1}{2p+2} - \epsilon$ approximation ratio, under time complexity of $\mathcal{O}(\frac{n}{\epsilon} \log \frac{r}{\epsilon})$.*

***Proof of Theorem 3***: We first consider the special case that $S_1$ or $S_2$ is empty, and show that TwinGreedyFast achieves $1 - \epsilon$ approximation ratio under this case. Without loss of generality, we assume $S_2$ is empty. By similar reasoning with the proof of Theorem 1 (Appendix A.3), we get $f(S_1 \mid \emptyset) \geq f(O \cap S_1 \mid \emptyset)$. Besides, for each $e \in O \backslash S_1$, we must have $f(e \mid \emptyset) < \tau_{min}$ (where $\tau_{min}$ is the smallest threshold tested by the algorithm), because otherwise $e$ should be added into $S_2$ by the TwinGreedyFast algorithm. By the submodularity of $f(\cdot)$, we have

$$
\begin{aligned}
f(O) - f(\emptyset) &\leq f(O \cap S_1 \mid \emptyset) + f(O \backslash S_1 \mid \emptyset) \leq f(S_1 \mid \emptyset) + \sum_{e \in O \backslash S_1} f(e \mid \emptyset) \\
&\leq f(S_1 \mid \emptyset) + r \cdot \tau_{min} \leq f(S_1 \mid \emptyset) + r \cdot \frac{\epsilon \cdot \tau_{max}}{r} \leq f(S_1 \mid \emptyset) + \epsilon f(O),
\end{aligned}
$$

which proves that $S_1$ has a $1 - \epsilon$ approximation ratio. In the sequel, we consider the case that $S_1 \neq \emptyset$ and $S_2 \neq \emptyset$. Let $O_5 = O \backslash (S_1 \cup S_2 \cup O_3)$ and $O_6 = O \backslash (S_1 \cup S_2 \cup O_4)$. By submodularity, we have

$$f(O \cup S_1) - f(S_1) \leq f(O_2^+ \mid S_1) + f(O_2^- \mid S_1) + f(O_3 \mid S_1) + f(O_5 \mid S_1) \tag{31}$$

$$f(O \cup S_2) - f(S_2) \leq f(O_1^+ \mid S_2) + f(O_1^- \mid S_2) + f(O_4 \mid S_2) + f(O_6 \mid S_2) \tag{32}$$

Using Lemma 3, we get

$$
\begin{aligned}
&f(O_2^+ \mid S_1) + f(O_2^- \mid S_1) + f(O_3 \mid S_1) + f(O_1^+ \mid S_2) + f(O_1^- \mid S_2) + f(O_4 \mid S_2) \\
&\leq (1 + \epsilon) \left[ \sum_{e \in O_1^+ \cup O_2^- \cup O_3} \delta(\pi_1(e)) + \sum_{e \in O_1^- \cup O_2^+ \cup O_4} \delta(\pi_2(e)) \right] \\
&\leq (1 + \epsilon) \left[ \sum_{e \in S_1} |\pi_1^{-1}(e)| \cdot \delta(e) + \sum_{e \in S_2} |\pi_2^{-1}(e)| \cdot \delta(e) \right] \\
&\leq (1 + \epsilon)p \left[ \sum_{e \in S_1} \delta(e) + \sum_{e \in S_2} \delta(e) \right] \\
&\leq (1 + \epsilon)p \left[ f(S_1) + f(S_2) \right], \tag{33}
\end{aligned}
$$

where the third inequality is due to Lemma 10. Besides, according to the definition of $O_5$, we must have $f(e \mid S_1) < \tau_{min}$ for each $e \in O_5$, where $\tau_{min}$ is the smallest threshold tested by the algorithm, because otherwise $e$ should be added into $S_1$ as $S_1 \cup \{e\} \in \mathcal{I}$. Similarly, we get $f(e \mid S_2) < \tau_{min}$ for each $e \in O_6$. Therefore, we have

$$f(O_5 \mid S_1) \leq \sum_{e \in O_5} f(e \mid S_1) \leq r \cdot \tau_{min} \leq r \cdot \frac{\epsilon \cdot \tau_{max}}{r} \leq \epsilon \cdot f(O) \tag{34}$$

$$f(O_6 \mid S_2) \leq \sum_{e \in O_6} f(e \mid S_2) \leq r \cdot \tau_{min} \leq r \cdot \frac{\epsilon \cdot \tau_{max}}{r} \leq \epsilon \cdot f(O) \tag{35}$$

As $f(\cdot)$ is a non-negative submodular function and $S_1 \cap S_2 = \emptyset$, we have

$$f(O) \leq f(O) + f(O \cup S_1 \cup S_2) \leq f(O \cup S_1) + f(O \cup S_2) \tag{36}$$

By summing up Eqn. (31)-(36) and simplifying, we get

$$
\begin{aligned}
f(O) &\leq [1 + (1 + \epsilon)p][f(S_1) + f(S_2)] + 2\epsilon \cdot f(O) \\
&\leq (2p + 2 + 2p\epsilon)f(S^*) + 2\epsilon \cdot f(O)
\end{aligned}
$$

So we have $f(S^*) \geq \frac{1-2\epsilon}{2p+2+2p\epsilon} f(O) \geq (\frac{1}{2p+2} - \epsilon)f(O)$. Note that the TwinGreedyFast algorithm has at most $\mathcal{O}(\log_{1+\epsilon} \frac{r}{\epsilon})$ iterations with $\mathcal{O}(n)$ time complexity in each iteration. Therefore, the total time complexity is $\mathcal{O}(\frac{n}{\epsilon} \log \frac{r}{\epsilon})$, which completes the proof. $\qquad\square$

## Appendix C: Supplementary Materials on Experiments

### C.1   Social Network Monitoring

It can be easily verified that the social network monitoring problem considered in Section 5 is a non-monotone submodular maximization problem subject to a partition matroid constraint. We provide additional experimental results on Barabasi-Albert (BA) random graphs, as shown in Fig. 3. In Fig. 3, we generate a BA graph with 10,000 nodes and $m_0 = m = 100$, and set $h = 5$ for Fig. 3(a)-(b) and set $h = 10$ for Fig. 3(c)-(d), respectively. The other settings in Fig. 3 are the same with those for ER random graph in Section 5. It can be seen that the experimental results in Fig. 3 are qualitatively similar to those on the ER random graph, and TwinGreedyFast still runs more than an order of magnitude faster than the other three algorithms. Besides, it is observed from Fig. 3 that TwinGreedyFast and TwinGreedy perform closely to Fantom and slightly outperform RRG and SampleGreedy on utility, while it is also possible that TwinGreedyFast/TwinGreedy can outperform Fantom on utility in some cases.

(a) BA, Number of Queries (h=5)

(b) BA, Utility (h=5)

(c) BA, Number of Queries (h=10)

(d) BA, Utility (h=10)

Figure 3: Experimental results for social network monitoring on Barabasi-Albert (BA) random graph

In Table 2, we study how the utility of TwinGreedyFast can be affected by the parameter $\epsilon$. The experimental results in Table 2 reveal that, the utility of TwinGreedyFast slightly increases when

Table 2: The utility of TwinGreedyFast ($\times 10^5$) vs. the parameter $\epsilon$ (BA, $h = 5$)

| $\epsilon$ | $k =$50 | 100 | 150 | 200 | 250 | 300 | 350 | 400 | 450 | 500 |
|---|---|---|---|---|---|---|---|---|---|---|
| 0.2 | 0.145 | 0.281 | 0.410 | 0.529 | 0.637 | 0.744 | 0.836 | 0.925 | 1.004 | 1.078 |
| 0.15 | 0.149 | 0.289 | 0.415 | 0.535 | 0.643 | 0.747 | 0.841 | 0.929 | 1.008 | 1.082 |
| 0.1 | 0.154 | 0.291 | 0.418 | 0.538 | 0.648 | 0.751 | 0.846 | 0.933 | 1.013 | 1.085 |
| 0.05 | 0.154 | 0.293 | 0.421 | 0.540 | 0.651 | 0.753 | 0.848 | 0.935 | 1.015 | 1.087 |
| 0.02 | 0.155 | 0.294 | 0.422 | 0.541 | 0.652 | 0.754 | 0.849 | 0.936 | 1.015 | 1.087 |
| 0.01 | 0.155 | 0.294 | 0.422 | 0.541 | 0.652 | 0.754 | 0.849 | 0.936 | 1.015 | 1.087 |
| 0.005 | 0.155 | 0.294 | 0.422 | 0.541 | 0.652 | 0.754 | 0.849 | 0.936 | 1.015 | 1.088 |

$\epsilon$ decreases, and almost does not change when $\epsilon$ is sufficiently small (e.g., $\epsilon \leq 0.02$). Therefore, we would not suffer a great loss on utility by setting $\epsilon$ to a relatively large number in $(0, 1)$ for TwinGreedyFast.

## C.2  Multi-Product Viral Marketing

We first prove that the multi-product viral marketing application considered in Section 5 is an instance of the problem of non-monotone submodular maximization subject to a matroid constraint. Recall that we need to select $k$ seed nodes from a social network $G = (V, E)$ to promote $m$ products, and each node $u \in V$ can be selected as a seed for at most one product. These requirements can be modeled as a matroid constraint, as proved in the following lemma:

**Lemma 11** *Define the ground set $\mathcal{N} = V \times [m]$ and $\mathcal{I} = \{X \subseteq \mathcal{N} : |X| \leq k \wedge \forall u \in V : |X \cap \mathcal{N}_u| \leq 1\}$, where $\mathcal{N}_u \triangleq \{(u, i) : i \in [m]\}$ for any $u \in V$. Then $(\mathcal{N}, \mathcal{I})$ is a matroid.*

***Proof of Lemma 11***:    It is evident that $(\mathcal{N}, \mathcal{I})$ is an independence system. Next, we prove that it satisfies the exchange property. For any $X \in \mathcal{I}$ and $Y \in \mathcal{I}$ satisfying $|X| < |Y|$, there must exist certain $v \in V$ such that $|Y \cap \mathcal{N}_v| > |X \cap \mathcal{N}_v|$ (i.e., $|Y \cap \mathcal{N}_v| = 1$ and $|X \cap \mathcal{N}_v| = 0$), because otherwise we have $|X| = \sum_{u \in V} |X \cap \mathcal{N}_u| \geq \sum_{u \in V} |Y \cap \mathcal{N}_u| = |Y|$; contradicting $|X| < |Y|$. As $|X| < |Y| \leq k$, we can add the element in $Y \cap \mathcal{N}_v$ into $X$ without violating the feasibility of $\mathcal{I}$, which proves that $(\mathcal{N}, \mathcal{I})$ satisfies the exchange property of matroids. $\square$

Next, we prove that the objective function in multi-product viral marketing is a submodular function defined on $2^{\mathcal{N}}$:

**Lemma 12** *For any $S \subseteq \mathcal{N}$ and $S \neq \emptyset$, define*

$$f(S) = \sum_{i \in [m]} f_i(S_i) + \left( B - \sum_{i \in [m]} \sum_{v \in S_i} c(v) \right) \tag{37}$$

*where $S_i \triangleq \{u \mid (u, i) \in S\}$ and $f_i(\cdot)$ is a non-negative submodular function defined on $2^V$ (i.e., an influence spread function). We also define $f(\emptyset) = 0$. Then $f(\cdot)$ is a submodular function defined on $2^{\mathcal{N}}$.*

***Proof of Lemma 12***:    For any $S \subsetneq T \subseteq \mathcal{N}$ and any $x = (u, i) \in \mathcal{N} \backslash T$, we must have $u \notin S_i$ and $u \notin T_i$. So we get $f(x \mid T) = f_i(u \mid T_i) - c(u)$. If $S \neq \emptyset$, then we have $f(x \mid S) = f_i(u \mid S_i) - c(u)$ and hence $f(x \mid T) \leq f(x \mid S)$ due to $S_i \subseteq T_i$ and the submodularity of $f_i(\cdot)$. If $S = \emptyset$, then we also have $f(x \mid S) = f_i(u) + B - c(u) \geq f_i(u \mid T_i) - c(u) = f(x \mid T)$, which completes the proof. $\square$

As we set $B = m \sum_{u \in V} c(u)$, the objective function $f(\cdot)$ is also non-negative. Note that $f_i(A)$ denotes the total expected number of nodes in $V$ that can be activated by $A$ ($\forall A \subseteq V$) under the celebrated Independent Cascade (IC) Model [34]. As evaluating $f_i(A)$ for any given $A \subseteq V$ under the IC model is an NP-hard problem, we use the estimation method proposed in [7] to estimate $f_i(A)$, based on the concept of "Reverse Reachable Set" (RR-set). For completeness, we introduce this estimation method in the following:

Given a directed social network $G = (V, E)$ with each edge $(u, v)$ associated with a probability $p_{u,v}$, a random RR-set $R$ under the IC model is generated by: (1) remove each edge $(u, v) \in E$ independently with probability $1 - p_{u,v}$ and reverse $(u, v)$'s direction if it is not removed; (2) sample $v \in V$ uniformly at random and set $R$ as the set of nodes reachable from $v$ in the graph generated by the first step. Given a set $Z$ of random RR-sets, any $i \in [m]$ and any $A \subseteq V$, we define

$$\hat{f}_i(A) = \sum_{R \in Z} |V| \cdot \min\{1, |A \cap R|\}/|Z| \tag{38}$$

According to [7], $\hat{f}_i(A)$ is an unbiased estimation of $f_i(A)$, and $\hat{f}_i(\cdot)$ is also a non-negative monotone submodular function defined on $2^V$. Therefore, in our experiments, we generate a set $Z$ of one million random RR-sets and use $\hat{f}_i()$ to replace $f_i()$ in the objective function shown in Eqn. (37), which keeps $f(\cdot)$ as a non-negative submodular function defined on $2^{\mathcal{N}}$.