[Reviews · NeurIPS 2020]

Review 1

Summary and Contributions: Contribution: The paper gives a new deterministic algorithm for maximizing a non-monotone submodular function subject to a matroid constraint. The algorithm achieves a 1/4 approximation and it has a running time of O(n r), where n is the size of the ground set and r is the rank of the matroid. Using known techniques, one can speed up the algorithm to nearly-linear at a small loss in the approximation. Comparison to previous work: Previously, there were two main approaches for obtaining deterministic algorithms for the problem. The first approach, due to Lee et al., uses local search and it obtains a 1/4 - eps approximation but the running time is a much larger polynomial (at least n^4). Another approach, due to Gupta et al., uses achieves the same running time as the proposed algorithm but achieves a weaker approximation of 1/6. It is instructive to compare the algorithms of Gupta et al. and the proposed algorithm. The former algorithm first constructs a solution A by running the standard Greedy algorithm for monotone maximization on the entire ground set, then it constructs a second solution B by running the Greedy algorithm on the complement of A, and finally it constructs a third solution C by running an algorithm for unconstrained maximization on A, and returns the better of the three solutions A, B, C. In contrast, the proposed algorithm simultaneously constructs two disjoint solutions using the Greedy algorithm; if an element can be feasibly added to both solutions, it is added to the one with higher marginal gain. Using a very simple but careful analysis along the lines of the standard Greedy analysis, the paper shows that the algorithm achieves a 1/4 approximation. The work [11] gives a fast randomized algorithm RandomGreedy and show it achieves a 1/4 approximation, and the work [14] derandomizes the same algorithm. An approximation guarantee is not provided in [14] for non-monotone functions, it would be interesting to see whether such an analysis can be obtained. Compared to the derandomized RandomGreedy, the algorithm proposed in this work is faster by an r factor. Novelty: The algorithm is very simple and natural, but it is a novel contribution to the line of work devoted to obtaining fast deterministic algorithms for non-monotone functions. The analysis is very simple but insightful. === After the author feedback === I have read the author feedback. I agree with the authors that the randomized algorithms are not necessarily fast and there is a clear running time advantage for the proposed algorithm.

Strengths: The paper introduces a novel deterministic algorithm for submodular maximization with a matroid constraint. This is a topic of interest to the theory community and the result is nice and simple. The relevance to the ML community seems much more limited.

Weaknesses: The contribution is primarily of theoretical interest, since it focuses on deterministic algorithms and algorithms with matching or better approximations are known if randomization is allowed. There is a clear running time advantage for the proposed algorithm though.

Correctness: I checked the proofs in detail and I think they are correct.

Clarity: Yes.

Relation to Prior Work: Yes.

Reproducibility: Yes

Additional Feedback:


Review 2

Summary and Contributions: The authors consider the problem of maximizing a (non-monotone, non-negative) submodular function subject to a matroid constraint. They develop a deterministic algorithm TwinGreedy that achieves a 1/4 approximation, which is the state-of-the-art for deterministic algorithms. They then modify their algorithm to achieve a 1/4 - epsilon approximation in time O(n log r / epsilon), where r is the rank of the matroid. Their analysis can be extended to p-systems as well. The authors then consider how their algorithm performs in practice.

Strengths: The algorithm proposed is practical to implement, deterministic, and efficient while achieving a provable approximation guarantee. In my experience, deterministic algorithms are preferred when possible as they lead to more stable solutions -- even if they do not improve the objective value much over randomized algorithms. The algorithmic framework is novel to the best of my knowledge and may be able to be applied to other problems. The authors achieve good practical performance with their algorithm. They achieve an objective value very close to the baseline considered with about two orders of magnitude fewer function evaluations.

Weaknesses: I don't have any major issues with the paper. For Fantom, there are randomized linear time algorithms that achieve a 1/2 approximation for unconstrained submodular maximization. How would performance change if this was used instead of the 1/3 approximation algorithm used in the paper?

Correctness: I skimmed through the proof of the main theorem and it seems correct. The experimental section also seems sound.

Clarity: The related work, description of the algorithm, and description of the experiments are clear and well written. The bar graphs in the experiments are poorly designed. At first I thought SampleGreedy was better than TwinGreedy. The bars in the key go in a different direction than the bars in the graph. At first I rotated the bars in the key to align with the bars in the graph but after reading the text I realized this was wrong. Please change how the algorithms are distinguished. The use of O1, ..., O8 in the proof is confusing. I would prefer more descriptive symbols based on the relationship between the symbols. For example, O_1 and O_2 differ only in using $\in$ or $\notin$, O_1 and O_3 differ only in swapping 1 and 2. Perhaps using superscripts + and - can distinguish between O_1, O_2 and O_3, O_4 and subscripts can distinguish between O_1, O_3 and O_2, O_4.

Relation to Prior Work: The authors do a detailed comparison of related algorithms and how they compare in runtime, approximation guarantees, and deterministic/random.

Reproducibility: Yes

Additional Feedback: How does the performance of the FastTwinGreedy algorithm change as epsilon changes? Is it possible to outperform Fantom with a different choice of epsilon? How about with TwinGreedy? What is the intuition for why the approximation holds for TwinGreedy but not regular greedy? What happens if rather than two sets you have three or more sets competing? In the experiment section, is there any reason to suspect that wall clock time would differ from number of queries?


Review 3

Summary and Contributions: The paper gives two deterministic algorithms achieving roughly 1/4 approximations to submodular maximization under a matroid constraint, crucially without monotonicity. Crucially these algorithms are much more efficient than the state of the art. The algorithms also recover the known monotone 1/2 approximations (which is almost the state of the art approximation). The techniques used are a very clever idea of constructing two sets simultaneously which allows for comparison against the optimum.

Strengths: I enjoyed this paper a lot. I think the ideas are very clever and also very natural, and put together in an effective way. The proofs, while a bit tedious, are fairly straightforward and I think the ideas used to compare the two sets to the optimal are innovative and bring new directions to algorithm design in this area.

Weaknesses: I did feel that there could be more guiding text in general. It took several reads to understand what many of the lemmas are trying to do, even though the structure of the arguments themselves can be described in a few words. For instance, Lemma 1 is largely just an appeal to matroid basis exchange properties, and Lemma 2 is largely just using definitions and submodularity. The paper is understandably quite notation heavy, but I feel the names given to the sets can be a bit more descriptive than e.g. O_i for i = 1, 2, 3, 4, 5, 6, 7, 8. The proof of Lemma 1 is algorithmic and it might be helpful for the reader to actually see the algorithm. Finally, I had a (possibly somewhat mild) concern about the additional assumptions of non-negativity of the function f and that f(0) = 0. I saw that several of the prior works also make the assumption that f is non-negative -- is this necessary? As I am a bit unfamiliar with the literature, what algorithms / guarantees are known when this non-negativity assumption is lifted? I also couldn't see where the assumption f(0) = 0 was used, but it is possible I just missed something.

Correctness: I believe the claims are correct.

Clarity: The paper is fairly easy to follow, but as I stated above my main concern is that it is extremely notation heavy with a minimal amount of guiding text. That being said, I think a little bit of guiding text would significantly improve the readability (as the ideas and proof strategies themselves are very natural).

Relation to Prior Work: Yes.

Reproducibility: No

Additional Feedback:


Review 4

Summary and Contributions: This paper proposes a deterministic 1/4-approximation algorithm for non-monotone submodular maximization with a matroid constraint that runs in O(nr) time. Also, the authors accelerate the algorithm and extend to p-set sytem constraints.

Strengths: This paper proposes TwinGreedy algorithm that returns a 1/4-approximate solution for non-monotone submodular maximization with a matroid constraint and runs in O(nr) time, where n is the size of the ground set and r is the matroid rank. This problem has been attracting a lot of attention as an important problem on subdmoular maximization and many kinds of approximation algorithms have been designed. TwinGreedy algorithm is basically a standard greedy algorithm but keeps two solutions. Its analysis is quite non-trivial and, to my knowledge, is not similar to any existing method. By utilizing the thresholding method proposed by Badanidiyuru and Vondrak (2014), the authors also propose TwinGreedyFast, which is an accelerated version of TwinGreedy algorithm. It is just a straightforward application of the existing idea, but it is practically absolutely efficient as illustrated in the experiment section. These results are extended to p-set system constraints (it is called p-system in most of the literature). The approximation ratio is 1/(2p+2). Compared to the existing (1/(p+2sqrt(p)+3)-eps)-approximation algorithm with O((nr+r/eps)sqrt(p)) time, the proposed one is always faster and achieves better approximation for small p. Since p-set systems include many important constraints, this algorithm would become a standard approach to non-monotone submodular maximization problems.

Weaknesses: The result (deterministic 1/4-approximation in O(nr) time) is the fastest among all deterministic 1/4-approximation algorithms, but it is hard to say this result is ground-breaking. There already exists a deterministic 1/4-eps approximation algorithm by Lee et al. and a randomized 1/4-approximation algorithm that runs in O(nr) time by Feldman et al.

Correctness: I briefly checked all the proofs. They seem to be correct.

Clarity: Overall, this paper is well written. The proofs are clearly presented.

Relation to Prior Work: There are several existing algorithms for non-monotone submodular maximization with a matroid constraint. In my opinion, this paper sufficiently sumamrizes these results. Since the result by Buchbinder and Feldman [10] is the best randomized approximation for this problem, it would be helpful if the authors add it to Table 1 though it is mentioned in related work section.

Reproducibility: Yes

Additional Feedback: # Update after the author feedback The authors clearly responded to the questions raised in my review. I agree with the acceptance of this paper.

[Author Response · NeurIPS 2020]

We are grateful for the insightful comments of all reviewers. We will revise our manuscript carefully to address all
clarity issues indicated by the reviewers. In the following, we address the concerns of each reviewer in detail:

**Reviewer 1:** "If my understanding is correct...Compared to the derandomized RandomGreedy, the algorithm proposed
in this work is faster by an $r$ factor" The potential derandomization approach sounds interesting, even though it would
have higher time complexity than our approach. As it is unclear how to bridge the gap between the method in [14] and
the performance analysis in [11], we will check it in more detail and add a discussion on it to our manuscript.

"The contribution is primarily of theoretical interest, since it focuses on deterministic algorithms and fast algorithms
with better approximations are known if randomization is allowed" To the best of our knowledge, the fastest algorithm
with an approximation ratio better than $\frac{1}{4}$ is Algorithm 4 in [11], which is randomized and has a 0.283 ratio. This
algorithm's time complexity (regarding the queries to value and independence oracles or other general operations) is
larger than TwinGreedy by at least an additive factor of $\mathcal{O}(r^3)$ (due to its Line 4). As $r$ can be in the order of $\Theta(n)$,
there might exist a tradeoff between accuracy and efficiency. Compared to this algorithm, our approach also has the
following features. First, our algorithms can be directly used to address a more general $p$-set system constraint with
good performance bounds (especially for small $p$). Second, our algorithms can be accelerated to achieve nearly linear
running time, as shown in Section 4. We will add more discussions on [11] to our manuscript.

**Reviewer 2:** "For Fantom, there are...instead of the 1/3 approximation algorithm used in the paper?"We tested Fantom
again using the randomized algorithm in [12] with $1/2$ expected ratio. The experimental results are almost the same as
before, although Fantom has a larger $1/6$ ratio (in expectation) in such a case. We will revise Section 5 to clarify this.

"How does the performance of the FastTwinGreedy algorithm change as epsilon changes? Is it possible to...How about
with TwinGreedy?" In our experiments, the utility of TwinGreedyFast slightly increases when $\epsilon$ decreases, and almost
does not change when $\epsilon$ is sufficiently small (e.g., $\epsilon \leq 0.02$). It is also possible that the utilities of TwinGreedyFast
(with a small $\epsilon$) and TwinGreedy outperform Fantom on some data points (but not for every input), and we will add
more experimental results about this to our manuscript.

"What is the intuition for...two sets you have three or more sets competing?" If regular greedy is used, we can only get a
set $S$ satisfying $f(S) \geq f(S \cup O)/2$, which implies that $S$ has $\frac{1}{2}$ approximation ratio if $f(\cdot)$ is monotone. When $f(\cdot)$ is
non-monotone, we cannot use $f(S \cup O) \geq f(O)$, so we have to use $f(O \cup S_1) + f(O \cup S_2) \geq f(O)$ (i.e., Eqn. (10))
to derive the ratio. As two solution sets are sufficient for Eqn. (10) to upper-bound $f(O)$ based on submodularity of
$f(\cdot)$, introducing more competing sets would not help to improve the ratio but would cause extra time complexity.

"In the experiment section, is there any reason to suspect that wall clock time would differ from number of queries?"
The wall clock running time of the algorithms does differ from the number of queries, but the experimental results are
qualitatively similar: TwinGreedyFast is still faster than the other algorithms by more than an order of magnitude, as
most of the running time is spent on oracle queries. We will add more experimental results on this to our manuscript.

**Reviewer 3:** "I saw that several of the prior works also make the assumption that f is non-negative-is this necessary?"
It is generally believed that submodular functions may only be optimized under the non-negativity assumption, so
most studies in the literature have adopted this assumption. A good reference (and also a rare exception) for this is
"Submodular Maximization beyond Non-negativity: Guarantees, Fast Algorithms, and Applications. ICML 2019". In
our paper, we have used the non-negativity assumption in Eqn. (10), i.e., $f(O) + f(O \cup S_1 \cup S_2) \geq f(O)$.

"I also couldn't see where the assumption f(0) = 0 was used..." For simplicity, we have followed many related
studies (e.g.,[15][16][31]) to assume $f(\emptyset) = 0$. Actually, the $\frac{1}{4}$ ratio of TwinGreedy still holds when $f(\emptyset) > 0$,
by minor revisions to the proof of Theorem 1: (1) In Eqn. (8), change $f(S_1) + f(S_2)$ to $f(S_1) + f(S_2) - 2f(\emptyset)$,
as $\sum_{e \in S_i} \delta(e) = f(S_i \mid \emptyset)$ for $i = 1, 2$; (2) Revise the proof in Section A.3 as follows. Suppose that $S_1 \neq \emptyset$
and $S_2 = \emptyset$. According to the greedy rule of the algorithm, we have $f(O \cap S_1 \mid \emptyset) \leq \sum_{e \in O \cap S_1} f(e \mid \emptyset) \leq$
$\sum_{e \in O \cap S_1} \delta(e) \leq \sum_{e \in S_1} \delta(e) = f(S_1 \mid \emptyset)$ and $f(O \backslash S_1 \mid \emptyset) \leq \sum_{e \in O \backslash S_1} f(e \mid \emptyset) \leq 0$. Combining these with
$f(O \backslash S_1) + f(O \cap S_1) \geq f(O) + f(\emptyset)$, we get $f(S_1) \geq f(O)$. By similar minor revisions, it can also be shown that
the performance bounds of TwinGreedyFast still hold when $f(\emptyset) > 0$. We will revise our paper to clarify these points.

**Reviewer 4:** "There already exists a deterministic 1/4-eps approximation algorithm by Lee et al. and a randomized
1/4-approximation algorithm that runs in O(nr) time by Feldman et al." We agree. While matching the ratio of these
algorithms, our approach also has the following features. First, our algorithms are simple can be accelerated to achieve
nearly linear running time. Second, our algorithms can be directly used to address a more general $p$-set system constraint.
Third, our algorithms are "unified" and also perform well under the monotone case. For example, by similar reasoning
with Theorems 1-3, it can be seen that TwinGreedy achieves $\frac{1}{p+1}$ ratio for monotone $f(\cdot)$ under a $p$-set system constraint,
which is almost best possible [2]. Fourth, our algorithms are deterministic and can achieve more "stable" practical
performance than randomized algorithms (with much lower time complexity), which is corroborated by Section 5. We
will revise our manuscript based on the reviewer's comments to make our contributions more clear.

[Meta-Review · NeurIPS 2020]

All reviewers that the paper gives a good algorithm for a problem of interest to the community. The results a simple, but nice.